# Milestone in predicting core plasma turbulence: successful multi-channel validation of the gyrokinetic code GENE

Klara Höfler ®[1,2] ✉, Tobias Görler ®[1], Tim Happel ®[1], Carsten Lechte[3], Pedro Molina[1,4,5], Michael Bergmann[1], Rachel Bielajew[4], Garrard D. Conway[1], Pierre David[1], Severin S. Denk[4,6], Rainer Fischer ®[1], Pascale Hennequin[7], Frank Jenko ®[1], Rachael M. McDermott ®[1], Anne E. White[4], Ulrich Stroth[1,2] & the ASDEX Upgrade Team*

On the basis of several recent breakthroughs in fusion research, many activities have been launched around the world to develop fusion power plants on the fastest possible time scale. In this context, high-fidelity simulations of the plasma behavior on large supercomputers provide one of the main pathways to accelerating progress by guiding crucial design decisions. When it comes to determining the energy confinement time of a magnetic confinement fusion device, which is a key quantity of interest, gyrokinetic turbulence simulations are considered the approach of choice – but the question, whether they are really able to reliably predict the plasma behavior is still open. The present study addresses this important issue by means of careful comparisons between state-of-the-art gyrokinetic turbulence simulations with the GENE code and experimental observations in the ASDEX Upgrade tokamak for an unprecedented number of simultaneous plasma observables.

In recent years, fusion research has been able to achieve several breakthroughs, including new world records in the JET tokamak (most fusion energy produced in a sustained manner) and in the Wendelstein 7-X stellarator (highest triple product of ion density, ion temperature, and energy confinement time in a stellarator). Meanwhile, we also saw the first-ever demonstration of fusion ignition by Lawrence Livermore National Laboratory's National Ignition Facility (NIF). Motivated by these significant advances, ambitious projects have been launched across different parts of the world with the goal to develop fusion power plants as fast as possible. In this context, first-principles based simulations of the plasma dynamics on large supercomputers play a key role, since they help guide crucial design decisions, thus saving time and resources.

Notably, a future fusion power plant will need to operate at values of the triple product of ion density $n_i$, ion temperature $T_i$, and energy confinement time $\tau_E$ above a threshold of $3 \times 10^{21}$ m$^{-3}$ keV s, as set by the Lawson criterion. Given that present-day tokamak devices routinely achieve the desired values for $n_i$ and $T_i$, $\tau_E$ is a key quantity of interest in fusion research, measuring the energy loss of the plasma due to turbulent transport. The latter phenomenon is known to be driven by an array of microinstabilities driven by gradients in temperature and density. These instabilities eventually saturate due to nonlinear effects, resulting in a quasi-stationary turbulent state characterized by small-amplitude fluctuations of many plasma parameters.

[1]Max Planck Institute for Plasma Physics, Boltzmannstr. 2, Garching, Germany. [2]Technical University of Munich, TUM School of Natural Sciences, Physics Department, James-Franck-Str. 1, Garching, Germany. [3]University of Stuttgart, Institute of Interfacial Process Engineering and Plasma Technology, Pfaffenwaldring 31, Stuttgart, Germany. [4]Plasma Science and Fusion Center, Massachusetts Institute of Technology, 77 Massachusetts Ave, Cambridge, Massachusetts, USA. [5]Ecole Polytechnique Fédérale de Lausanne (EPFL), Swiss Plasma Center (SPC), Ecublens, Lausanne, Switzerland. [6]General Atomics, General Atomics Court, San Diego, California, USA. [7]Laboratoire de Physique des Plasmas, Ecole Polytechnique, Rte de Saclay, Palaiseau, France. *A list of authors and their affiliations appears at the end of the paper. ✉e-mail: klara.hoefler@ipp.mpg.de

During the last two decades or so, plasma turbulence simulations based on gyrokinetic models have made a remarkable leap forward. While initially, the focus was primarily on the understanding of the fundamental turbulence characteristics in idealized setups, nowadays, physically comprehensive simulations of the plasma behavior under realistic conditions are carried out on large supercomputers. Meanwhile, plasma diagnostics and the interpretation of experimental data have also evolved substantially, and this now allows a rigorous validation of the physics models used in the codes. Here, the latter must transcend global quantities, such as spatiotemporally averaged heat fluxes, and necessarily also take into account the characteristics of the underlying fluctuations, including amplitudes, wavenumber spectra, and cross phases[1].

Obviously, with every single feature being added to a quantitative comparison between simulation and experiment, the validation becomes more rigorous and convincing. We point out that a rigorous validation is an extensive effort, even for one single quantity, as it involves detailed assessments of systematic and random (measurement) errors[2,3], synthetic diagnostics, and the definition of a metric that quantifies the degree of agreement[4]. In order to facilitate such studies, the data from this study will be made available online for more in-depth validation work. In the present work, we focus on the best possible comparison of the largest number of experimental observables and their comparison to simulation results via synthetic diagnostics. The plasma scenarios were carefully designed, and the measurement diagnostics were pushed to their limits to gather comprehensive data at two radial positions in the core plasma. Since both positions yielded equally good results, only one is presented for clarity. While validating boundary plasma models is also essential, the remarkable agreement between our measurements and simulations in the core plasma represents a significant achievement and a key step towards advancing fusion power plant design.

Our study puts the ability of gyrokinetics to accurately describe the small-scale plasma dynamics to a hard test. Figure 1(a) lists several past works and indicates the turbulence observables used for the comparison with the simulation results[5–13]. The present study compares all of them between experiment and simulation: the characteristics of both density and temperature fluctuations, including their scale dependence and phase relationship were measured on ASDEX Upgrade (AUG)[14], one of the world's leading tokamak experiments. For two plasma scenarios with different electron temperature gradients, the experimental measurements are compared with simulation results from the state-of-the-art plasma turbulence code GENE[15,16]. Advanced synthetic diagnostics are used to obtain the most reliable comparisons. In this context, it is found that the observations are quantitatively reproduced by the simulations to a large degree of accuracy. We conclude that gyrokinetic codes have reached a high level of maturity, which allows them to be used as reliable tools to predict core plasma turbulence. This progress significantly contributes to advancing the design of future fusion power plants.

## Results

Comparisons of complex measurement data with simulation results must be carried out with the utmost care. Figure 1b illustrates the process applied here. An experiment is performed and diagnosed in the best possible way. The background quantities, such as the magnetic equilibrium and kinetic profiles, serve as input for the gyrokinetic turbulence simulations. The turbulent output fields generated by the code must be analyzed using synthetic diagnostics that replicate the underlying physical processes of the measurements, with a primary focus in this work on the microwave-plasma interaction. The output of the synthetic diagnostic measurements is then compared to their real-life counterparts that were measured in the experiment.

Figure 2 shows the experimental setup for the measurement of turbulence parameters. Panel (a) depicts the AUG cross-section, where solid (dashed) black lines indicate closed (open) magnetic flux surfaces. A snapshot of a turbulence simulation shows the relative amplitudes of the electron density fluctuation in color-coded form. Zooms to the main measurement region are shown in panels (b, c). The density fluctuations in panel (b) are presented together with a trajectory (green line) from ray-tracing calculations[17], which reproduces a path of the microwave beam launched by the Doppler backscattering diagnostic (DBS)[18] in this experiment. The weighting function (in grayscale) from two-dimensional full-wave analysis[19] indicates the probed plasma volume. Panel (c) shows the measurement volumes for a correlation electron cyclotron emission (CECE) diagnostic[20] that measures the electron temperature fluctuation amplitudes. The volumes have been obtained through analysis by the Torbeam[17,21] and ECRad codes[22].

The experimental data is from deuterium low-confinement mode (L-mode) discharges on the AUG tokamak (minor and major plasma radii are $a = 0.57$ m and $R_0 = 1.657$ m, respectively) in upper single null magnetic configuration (Fig. 2) to avoid transition to the high confinement mode (H-mode). L-mode is targeted in these experiments, since on AUG in H-mode, the normalized ion gyroradius, $\rho^*$, usually takes on values that violate the assumption for local gyrokinetic theory[23,24]. The magnetic field strength was $-2.537$ T, the plasma current 0.87 MA. The auxilliary heating power was 0.87 MW from neutral beam injection and 1.07 MW from electron cyclotron resonance heating (ECRH). With all other control parameters constant, the ECRH deposition radius of one 0.5 MW injector was varied to change the electron temperature ($T_e$) profile from steep to flat, as we will refer to the two cases. This was done to vary the turbulence drive. Together with the kinetic plasma parameter profiles, Fig. 3d shows the increase of the normalized $T_e$ gradient $R/L_{T_e}$, with the gradient length

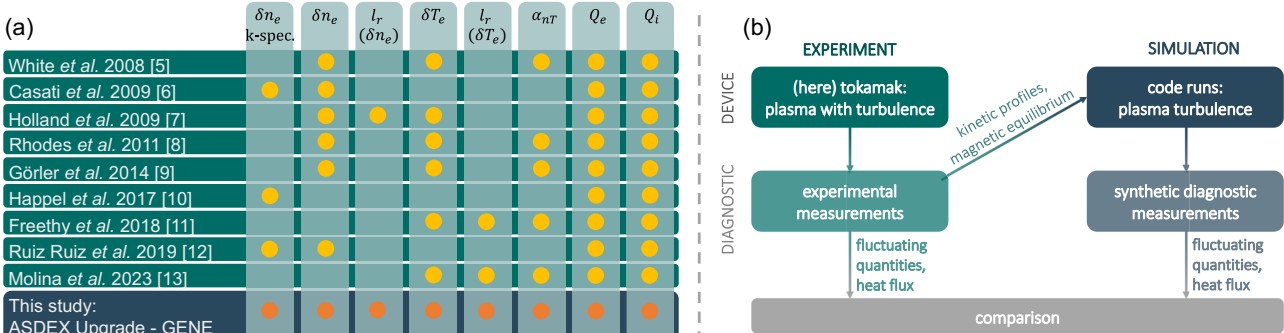

**Fig. 1 | Previous validation works and workflow. (a)** (Incomplete) list of previous validation works comparing several turbulence observables between experiment and gyrokinetic simulations[5–13]. The current study compares the to date largest number of turbulence quantities simultaneously. **b** Flow chart describing the steps involved in comparative studies. For details, refer to the text.

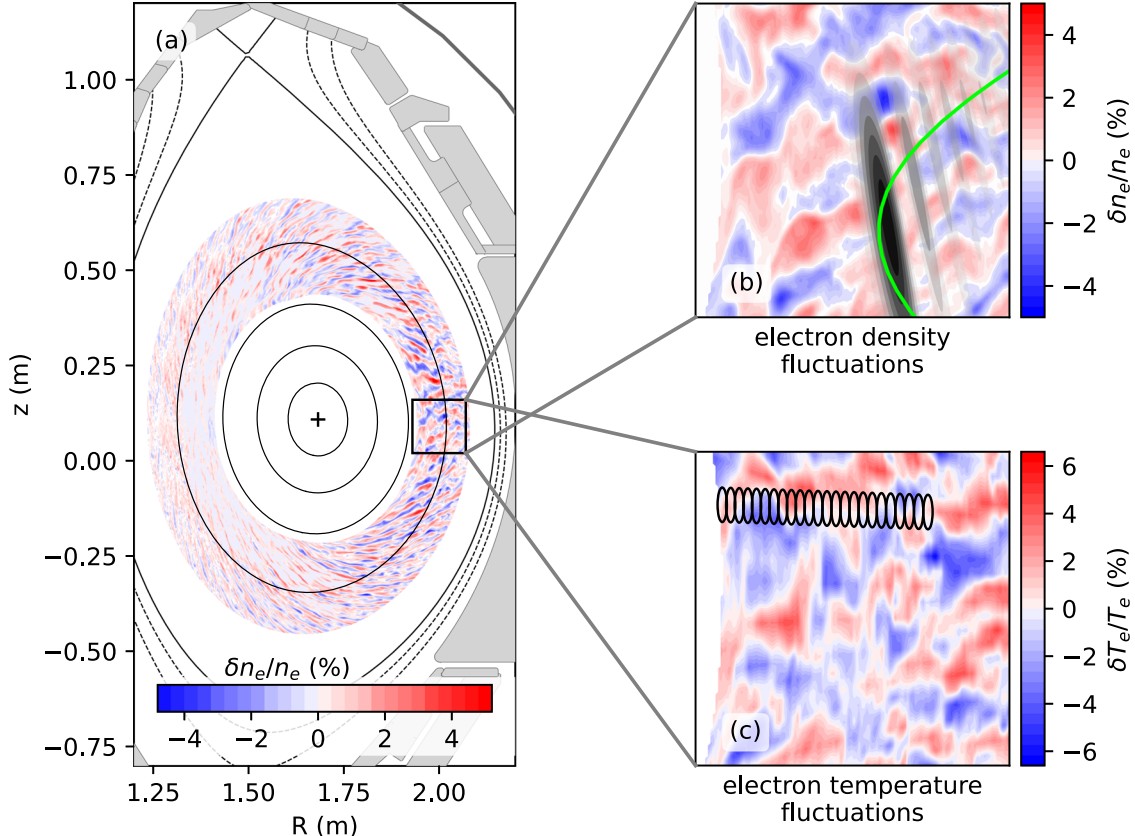

**Fig. 2 | Experimental set up. a** Poloidal cross-section of AUG including flux surfaces (black lines) and density fluctuations from the gyrokinetic simulation. The zoomed windows show (**b**) density fluctuations at the measurement position. Additionally, the probing beam from ray-tracing (green) and the weighting function from 2D full-wave simulations (shades of grey) are shown. Panel (**c**) depicts temperature fluctuations along with the CECE measurement volumes (black ellipses) obtained from the Torbeam and ECRad codes.

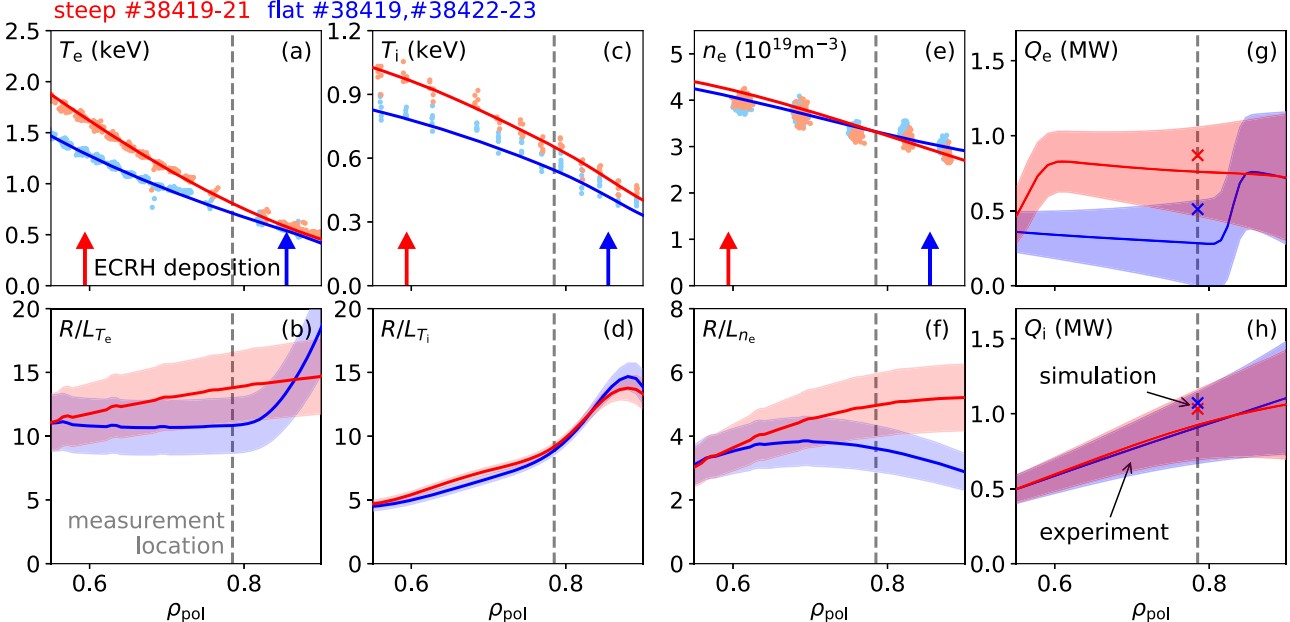

**Fig. 3 | Input to the gyrokinetic simulations and comparison of heat fluxes.** Profiles and normalized gradients of the electron temperature (**a**, **b**), ion temperature (**c**, **d**) and electron density (**e**, **f**). The points in (**a**, **c**, **e**) indicate experimental measurement data. The lines and uncertainties are output of Bayesian analysis in (**a**–**f**). Radial profiles of (**g**) electron and (**h**) ion surface-integrated heat fluxes: transport analysis results are shown as solid lines with uncertainty bands from Markov Chain Monte Carlo modeling, gyrokinetic simulation results are plotted as symbols x.

$L_{T_e} = \partial_r \ln T_e$, as the ECRH deposition radius moves from $\rho_{pol} \approx 0.85$ to $\rho_{pol} \approx 0.60$; the normalized radial coordinate increases from $\rho_{pol} = 0$ in the plasma center to 1 at the last closed flux surface.

The kinetic profiles in Fig. 3a–f were recorded during 400 ms long stationary discharge phases. For the comparison with the simulated heat fluxes later on, the quality of the fits is of particular importance. They were determined for the full radial region ($\rho_{pol} = 0.0$–1.2) by an integrated Bayesian data analysis using the IDA framework[25]. For the $T_e$ profile (Fig. 3a, b), data from an electron cyclotron emission (ECE) radiometer[26] and from a Thomson scattering diagnostic[27] were taken into account. The density ($n_e$) profiles (Fig. 3e, f) rely on Thomson scattering and interferometer[28] data, and the ion temperature ($T_i$) on active charge exchange recombination spectroscopy[29,30]. Consistent with prior studies[18] uncertainties of 20% in the normalized gradients of the electron domain are plausible (shaded areas in b, f). For the normalized gradient of $T_i$ the uncertainties are calculated with Gaussian processes regression. With the change from the flat to the steep scenario, $T_i$ and its normalized gradient (Fig. 3c, d) remain relatively unchanged, but the density profile steepens (Fig. 3e, f). This peaking of the density profile is attributed to a change in collisionality in a turbulence regime where both the ion-temperature gradient (ITG) and the trapped electron mode (TEM) are active[31]. The turbulence measurements were obtained at two radial positions, $\rho_{pol} = 0.74$ and 0.79; here we focus on the outer radius.

Scale-resolved density fluctuations were measured with three Doppler backscattering systems simultaneously, two in O-mode and one in X-mode polarization[10,18,32]. In order to obtain density fluctuation wavenumber ($k$) spectra at different radial positions, data were taken at a large number of probing frequencies and mirror angles, which direct the microwave to the desired poloidal position and sets the $k_\perp$ value of the measurement, where $k_\perp$ is the wavenumber perpendicular to the radial direction and to the magnetic field. A wavenumber range of $k_\perp = 3$–15 cm$^{-1}$ was measured. Electron temperature fluctuations were probed at $k_\perp \lesssim 4$ cm$^{-1}$ with the CECE system[20,33] with 200 MHz bandpass filters. Due to a perpendicular line of sight, the diagnostic is sensitive to large-scale fluctuations. For the $n_e$-$T_e$cross-phase measurements, one of the reflectometers was set to perpendicular incidence, where it probes the large scales, too, while viewing the same volume as the respective CECE channels.

The set of experimental data is compared with results from linear and nonlinear gyrokinetic simulations performed with the GENE code[15,16]. Being among the leading gyrokinetic codes worldwide, it is massively parallelized. Throughout its more than two decades of development, it has been successfully compared and benchmarked with other comparable codes, e.g., Refs. 9,24,34,35, such that conclusions from this work are to some extent also applicable to other gyrokinetic codes. GENE allows us to consider electromagnetic fluctuations, inter- and intra-species collisions, external $E \times B$ shear flows and an arbitrary number of species in arbitrarily shaped flux surfaces either in radially global or local flux-tube simulation domains. Due to the small gyroradius-to-machine-size ratio at the radial positions of interest (c.f. Supplementary Tab. 1), the latter is employed in the present cases to benefit from the higher accuracy of the underlying spectral methods. Furthermore, the finer electron gyroradius scales have been excluded from nonlinear simulations, since they have been found to be negligible. Also, all experimental measurements have been performed on ion gyroradius scales. It should be noted that the influence of small-scale turbulence in the electron-temperature-gradient (ETG) driven range and its cross-scale coupling to larger scales has been observed in simulations and might be important for some scenarios[36,37]. Two species, electrons and deuterium ions, are considered. Impurities are introduced through the effective charge $Z_{eff}$ in the collision operator.

The experimental kinetic profiles in Fig. 3a–f, the measured background shear flow, and a pressure constrained magnetic equilibrium coupled with current diffusion[38] serve as input for the code. In the simulations, the normalized temperature and density gradients were varied within the experimental error bars to achieve the best possible agreement with the experimental surface-integrated heat flux. Figure 3g, h shows the resulting electron (g) and ion (h) surface-integrated heat fluxes in comparison with their experimental counterparts which come from transport analyses with the ASTRA code[39] solving the power balance equations. By varying the normalized gradients within experimental error bars, GENE finds surface-integrated heat fluxes close to the experimental values for both scenarios. With matched surface-integrated heat fluxes in both the ion and the electron domain, the analysis now turns to the comparison of code results with experimental fluctuations measurements.

## Electron temperature fluctuation amplitudes

The comparison starts with electron temperature fluctuation frequency spectra. An increase in the fluctuation amplitude $\tilde{T}_e$ is to be expected at the transition from flat to steep temperature profiles, where the local surface-integrated heat flux rises by around 0.5 MW (cf. Fig. 3g). The experimental power spectra in Fig. 4 (solid lines) for the steep (a) and the flat (b) scenario confirm this expectation. The simulated spectral power densities (dashed lines) reproduce this trend, and achieve good quantitative agreement. We stress that the y-axis has not been scaled and is the same for experiment and simulation. The experimental and simulated absolute values of the relative $T_e$ fluctuation amplitude are also indicated in the plots. They agree remarkably well in the drop from $\tilde{T}_e/T_e \approx 1\%$ in the steep scenario to 0.6 % in the flat one. The experimental spectra are shifted to slightly higher frequencies with respect the simulated ones. This is most likely due to difficulties in the experimental determination of the radial electric field and its shear having an impact on the shape of the CECE spectra, as presented in[13]. The overall shapes of the spectra agree well in both scenarios.

The result from this section is consistent with the conjecture that a steeper normalized temperature gradient results in a higher fluctuation level that is responsible for the increase in surface-integrated heat flux[40,41].

## Electron density fluctuation amplitudes

Next, spatial scale-resolved electron density fluctuations, $\tilde{n}_e$, are investigated for the two scenarios. Since the seminal works on 3D[42] and, for magnetized plasma more relevant, 2D[43] fluid turbulence, the $k$-spectra are known to contain information about the driving instabilities, the nonlinear energy cascades, and the dissipation mechanisms of turbulence. Experimental $k$-spectra from fusion plasmas are scarce. In the range where the drive of ITG and TEM turbulence is expected to happen, at $k \rho_s \approx 0.2$–0.3, with $\rho_s \propto \sqrt{T_e/m_i}$ being the ion drift scale, DBS is best suited to measure local density fluctuation spectra[10,44,45]. First comparisons of experimental and simulated $k_\perp$-spectra have shown the importance of using synthetic diagnostics to reach agreement[10,46].

Figure 4c–e presents the $k_\perp$-spectra compiled at $\rho_{pol} = 0.79$, in a volume between the two ECRH locations, for the flat and the steep plasma scenario (cf. Fig. 2). Turbulence was probed in O-mode by two DBS systems (Fig. 4c) and X-mode (Fig. 4d) polarization by one DBS system. For better visibility, individual vertical offsets were applied to the spectra pairs, since the applied technique only allows for the experimental measurement of relative fluctuation amplitudes. The offsets between the flat and steep scenarios in each spectral pair, however, are deterministic. The agreement of the shapes of the spectra measured by the two O-mode DBS systems is very satisfactory for both plasma scenarios. They have the typical shape of a 2D turbulence spectrum, however, because of the diagnostic response, it is inherently different from that of the actual density fluctuation spectrum as obtained from the GENE simulations (Fig. 4e). Therefore, a direct comparison is not appropriate. Due to the stronger microwave-plasma interaction of

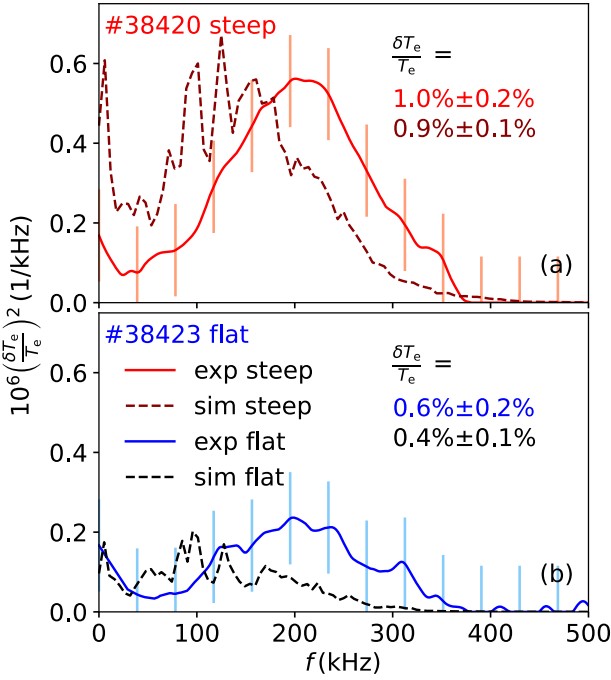

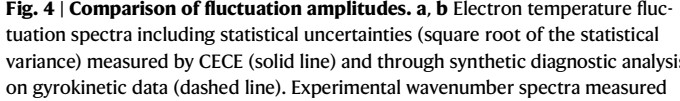

**Fig. 4 | Comparison of fluctuation amplitudes. a, b** Electron temperature fluctuation spectra including statistical uncertainties (square root of the statistical variance) measured by CECE (solid line) and through synthetic diagnostic analysis on gyrokinetic data (dashed line). Experimental wavenumber spectra measured with DBS ((**c**) two in O-mode, (**d**) one in X-mode) for the flat (blue) and steep (red) scenarios and comparison with two-dimensional full-wave simulation (black, dark red). **e** Wavenumber spectra calculated directly from the turbulence field from the gyrokinetic simulations.

X-mode waves[47], the corresponding spectra (Fig. 4d) have a different form again.

Figure 4c, d additionally show the synthetic diagnostic spectra relevant for the comparison. They come from the full-wave code IPF-FD3D, which determines the DBS responses of O- and X-mode waves scattered on the simulated density fluctuations. The synthetic spectra agree in shape with their experimental counterparts. This observation is in line with initial comparison studies on similar discharges[10,47].

An unexpected new observation is that the electron density fluctuation amplitude is higher in the flat scenario, where the density gradient is steeper by about 40 % than in the steep temperature gradient scenario (cf. Fig. 3f). This trend, which is evident in both the experimental and the synthetic spectra, contradicts the mixing length argument,

$$\tilde{n}/n \sim 1/(k_\perp L_n),\qquad(1)$$

which predicts an increase of the density fluctuation amplitude with density gradient. The observed difference is largest for intermediate scales, whereas at minimum and maximum $k$-values the fluctuation amplitudes become comparable. Already the raw spectrum in Fig. 4e, obtained from the simulated density fluctuations, shows higher fluctuation amplitudes for the flat scenario, showing that this is not a diagnostic effect.

It is remarkable that a concomitant increase in $\tilde{T}_e$ and decrease in $\tilde{n}_e$, as observed in the experiment, is reproduced by the gyrokinetic simulation. The linear growth rates (shown in Supplementary Fig. 1) are higher for the steep scenario, and as such, should result in higher fluctuation levels. The nonlinear response of the density field results in a density fluctuation energy decrease, though.

**Cross-phase between electron temperature and density fluctuations**

The cross-phase between electron density and temperature fluctuations is indicative of the nature of the microinstability providing energy for the turbulence[48]. In the experiment, it is estimated from a correlation analysis of a CECE channel with a reflectometer signal probing the same plasma volume. Previous studies on AUG correlated the amplitude of the reflectometer signal with the ECE signal[11,13,49], whereas studies on the DIII-D tokamak used also the reflectometer phase signal[50]. The present study uses the amplitude signal from an O-mode reflectometer, since it produces a higher coherence and thus more significant cross-phase measurements, as it was also reported from DIII-D[50].

Figure 5 compares the experimental and the simulated cross-phases showing no or little change from the flat to the steep scenario in the frequency range of the coherence being above the statistical limit (strong blue and red color). This indicates that the increase in temperature gradient is not sufficient to modify substantially the microinstability that feeds energy into the turbulence. However, there are differences between the experiment and the simulation both in absolute value and in the fact that the experimental value decreases with frequency. This ramping can be explained by a poloidal offset in CECE and reflectometer measurement positions of 2.5 cm. Due to differences in refraction when probing density and temperature fluctuations (the probing frequencies differ by a factor of about 2), a poloidal offset of at least 1.5 cm is expected. The remaining offset can be a consequence of uncertainties in the magnetic equilibrium reconstruction, which is more challenging in upper single null. Ramping phase angles due to a poloidal misalignment of CECE and reflectometer channels have previously been observed in synthetic diagnostic modeling on turbulence from another gyrokinetic code (GYRO), based on experimental measurements from DIII-D[51].

**Radial correlation length of electron density fluctuations**

The perpendicular wavenumber spectra in Section "Electron density fluctuation amplitudes" addressed the spatial extent of the density fluctuations in the binormal direction, i.e., perpendicular to the magnetic field and the radial direction. In this section, their radial extent is analysed, which is expressed in terms of the radial correlation length. It is deduced from two DBS channels, where one channel scans the radial

range around the measurement position of the other (reference) channel[52]. Since DBS is sensitive to fluctuations at a particular perpendicular scale, the radial correlation length is studied as function of the perpendicular density fluctuation scale length. This was done by repeating the measurment procedure for different DBS probing angles. The gyrokinetic data is again probed via 2D full-wave simulations; the analysis is done as in the experiment.

Figure 6a presents experimental and simulated radial correlation lengths of density fluctuations, $l_{r,n_e}$, as function of the perpendicular scale, $k_\perp$. As before, experiment (solid points) and simulation (symbol x) show a remarkable agreement, not only in the trend between steep and flat scenarios (red vs. blue), but also in terms of absolute values. Fits with a $1/k_\perp$ dependence are shown to guide the eye. Setting $l_\perp = \lambda_\perp / 2 = \pi/k_\perp$ yields $l_\perp/l_r \approx 1.5$, which means that structures are about 50% larger in the perpendicular (almost poloidal) direction than in the radial direction. This is indicative of the effect of velocity and magnetic shear, which can limit the radial structure size.

### Radial correlation length of electron temperature fluctuations

The radial correlation length of electron temperature fluctuations, $l_{r,T_e}$, results from a coherence analysis between a reference and several radially neighbouring ECE channels. The analysis follows the procedure introduced in a parallel study on AUG[13]. Figure 6b depicts the resulting cross-correlation as function of the channel separation. In the steep scenario (Fig. 6b), the width of Gaussian fits yields values of $l_{r,T_e} = 9$ mm and 12.7 mm in the simulation and the experiment,

respectively, which is a reasonable agreement. In case of the flat scenario, the simulation yields similar $l_{r,T_e}$ values as in the steep case. The experimental data lie on the Gaussian fit to the simulation data, but are insufficient to calculate a correlation length.

Seen in the light of turbulent eddies mixing the background profiles, one would expect similar values for the correlation lengths of the temperature and density fluctuations. This is indeed the case: The CECE measurements are sensitive to scales up to $k_\perp \lesssim 4.1$ cm$^{-1}$ in the steep scenario and $k_\perp \lesssim 3.9$ cm$^{-1}$ in the flat scenario. Thus, the correlation length of $L_{r,T_e} \approx 1$ cm is compared with $l_r$ from the low $k_\perp$ measurements from DBS in Fig. 6a, which are of very similar size. To our knowledge, this is the first comparison between experiment and simulation on core radial correlation lengths in both electron density and electron temperature. The finding of similar electron temperature fluctuation correlation lengths for the steep and flat scenario supports earlier simulation results that studied the dependence of $l_{r,T_e}$ on the surface-integrated electron heat flux[53].

To sum up this work, an unprecedented number of measured background and fluctuation parameters were compared with simulation results in two different plasma scenarios. The high level of agreement in all parameters is impressive. The fact that agreement was achieved in both plasma scenarios, with flat and steep electron temperature gradients, further strengthens the positive outcome of this study.

In the past, the design of fusion power plants was projected on the basis of purely empirical scaling laws for the energy confinement time. To obtain more detailed and scientifically sound information about the performance of a fusion power plant, reliable prediction of the kinetic profiles is essential. For the prediction of core turbulence, gyrokinetic codes are the favored tools. They are also used to derive transport coefficients for integrated modelling[54,55] of entire plasma discharges. This underlines the importance of the work presented here.

After a quarter of a century of continuous development, gyrokinetic codes can be used to predict the performance of the core plasma. The simulated scale-resolved turbulence characteristics down to millimeter scales agree strikingly well with experimental observations for both temperature and density fluctuations. The value of this result is further strengthened by the excellent agreement for macroscopic quantities that arise from these microscale turbulence characteristics. The present work thus creates the necessary confidence in the codes so that fusion researchers can rely on them when planning future tokamak experiments or fusion power plants.

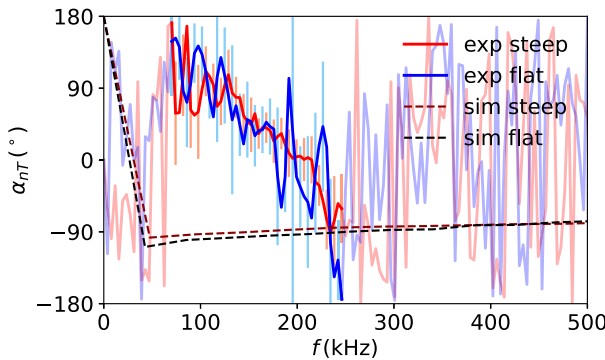

**Fig. 5 | Comparison of cross-phase $\alpha_{nT}$ between density and electron temperature fluctuations.** The frequency region with significant coherence between the $n_e$ and $T_e$ signals is shown in stronger colors than regions with no coherence. The error bars indicate the statistical uncertainty (square root of the statistical variance) For more details, refer to the text.

### Methods

Details of all applied methods can be found in ref. 56 and references therein. The key tools are summarized in the following.

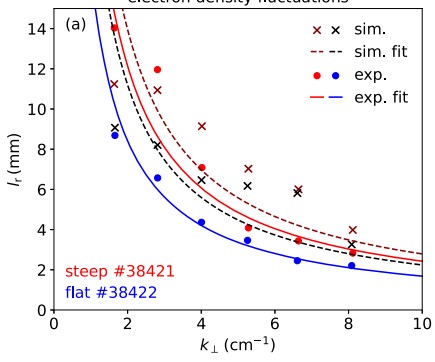
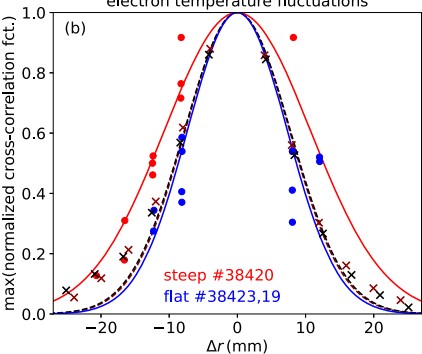

**Fig. 6 | Comparison of radial correlation lengths. a** Radial correlation length of electron density fluctuation $l_{r,n_e}$ versus inverse perpendicular structure size $k_\perp$ for both steep (red) and flat (blue) scenarios. Experimental data (full symbols) and fits (solid lines) compared to synthetic diagnostic analyses of gyrokinetic simulations (symbol x) and fits (dashed lines). **b** Maximum of the normalized cross-correlation between adjacent CECE channels (circles) and between simulated electron temperature fluctuations averaged in the corresponding plasma volumes (symbols x) for the steep (red) and flat (blue) scenario.

## Experimental measurements

Both plasma scenarios of this study are low confinement mode discharges in upper single null magnetic configuration with a plasma current of 0.8 MA, an on-axis magnetic field of $-2.53$ T, auxiliary neutral beam heating of 0.8 MW, 0.5 MW of central ECRH and 0.5 MW ECRH at two different radial locations.

The kinetic quantities $T_e$, $n_e$ and $T_i$ shown in Fig. 3 and the toroidal rotation velocity $v_{tor}$ serve as input to a pressure constrained magnetic equilibrium[38]. In this study, the Bayesian integrated data analysis (IDA) is used for reconstructing the profiles of $T_e$ and $n_e$ with a reduced number of spline knots compared to Ref. 25 to allow less spatial variation in the gradients. The profiles of $T_i$ and $v_{tor}$ are fitted on experimental charge exchange recombination spectroscopy data[29] using Gaussian process regression[57].

The one dimensional profile of plasma radiation is reconstructed tomographically from measurements of an array of bolometers[58].

The experimental surface-integrated heat fluxes of the electrons and ions are reconstructed using the ASTRA code[39] in interpretative mode.

**Measurement and analysis of electron density fluctuations.** Three DBS channels measure $\tilde{n}_e$ using the same steerable mirror in sector 11 of ASDEX Upgrade[59]. Two of them probe in the V-band frequency range in O-mode, one probes in the W-band frequency range in X-mode.

The fluctuation amplitude of $\tilde{n}_e$ is taken as the integral of a Gaussian fit to the asymmetric part of the power spectrum of the heterodyne backscattered signal[32]. To obtain wavenumber spectra, the probed wavenumber $k_\perp$ is varied by slowly steering the launching mirror ($\approx$2 deg/s) from perpendicular incidence to its maximum deflection. Meanwhile the probing beam frequency steps in plateaus of 5 ms each.

The radial correlation length analysis using the two O-mode DBS systems closely follows the procedure described in ref. 60. In this study the hopping channel probes at 20 different frequency plateaus lasting 10 ms each. The radial correlation length is the half width half max of a Lorentzian fit to the maxima of the cross-correlation functions versus the radial channel separation. The DBS signals used for cross-correlation are filtered in frequency space to remove noise.

For cross-phase measurements the launching mirror is operated in perpendicular incidence, since in this experiment the DBS systems share their line of sight with a CECE radiometer (see below). The length of one frequency plateaus of the reflectometers is 1 s. This paper shows an O-mode reflectometer measurement correlated with CECE.

**Measurement and analysis of electron temperature fluctuations.** A CECE radiometer probing in F-band frequency range[20,33] measures $\tilde{T}_e$ in sector 9 of ASDEX Upgrade.

Both the analysis of fluctuation amplitudes and radial correlation lengths uses time windows of steady state plasma conditions with a length of 4 s to reduce thermal noise when correlating different CECE channel pairs. The analysis of radial correlation lengths follows the methology described in Ref. 13, in which the thermal noise contribution is analytically excluded.

For measurements of the cross-phase between $n_e$ and $T_e$ the CECE system is moved to sector 11. There it shares the line of sight and data acquisition system with the DBS systems, which turn into reflectometers by probing in perpendicular incidence.

## Simulation

All analyses (gyrokinetic code GENE, full-wave code IPF-FD3D, beam-tracing code Torbeam, transport code ASTRA) consistently use the same kinetic profiles (cf. Fig. 3(a–f)) and pressure constraint equilibrium as input. The input quantities are listed in Supplementary Tab. 1.

**Turbulence simulations.** The primary tool employed for investigating the underlying plasma turbulence is the highly parallelized GENE

code[15,61]. As a Eulerian code, GENE implements the electromagnetic Vlasov-Maxwell equations on a fixed grid, allowing for flexibility in simulation domains from local to radially global. The simulations in this study, adopting the local approximation, benefit from periodic boundary conditions perpendicular to magnetic field lines, a computationally efficient choice, especially for larger machines like AUG. This approach proves effective when the ion-gyro-radius is small compared to the machine size, resulting in significant computational time savings. Consequently, the simulations encompass a diverse physics landscape, incorporating electromagnetic fluctuations, flux surface shaping via field-line tracing[62], linearized Landau-Boltzmann collisions (among other choices), external shear flow effects, and - in some cases multiple - fully gyrokinetically treated ion and electrons species with realistic mass ratios.

To further optimize the simulation costs, GENE employs field-aligned coordinates and splits the distribution function into a static background and so-called delta-f fluctuations. The default mode of operation is hence gradient-driven, i.e. evaluating the turbulence response for given and - on average - static temperature and density profiles and gradients informed by experimental measurements. This coincides nicely with the application targeted in this paper but experimental uncertainties necessitate corresponding parameter scans as presented above. While GENE accommodates alternative choices[63], a Maxwellian background distribution in velocity space is assumed, reflecting the thermalized nature of species.

Finally, Large-Eddy-Simulation techniques[64] are employed to ideally adjust the dissipation at the highest resolved wave-numbers.

This concise methodology ensures that GENE simulations remain versatile and robust, aligning with the intricacies of real-world plasma turbulence.

**Complementary results from linear turbulence simulations.** Linear GENE runs identify the dominant microinstabilities, which drive turbulence at different scales. Supplementary Fig. 1 shows the growth rates in (a) and real mode frequencies in (b),(c) of the fastest growing modes versus the inverse eddy scale size, $k_\perp$. Positive frequencies correspond to ITG-dominated turbulence, which moves in ion diamagnetic direction and negative frequencies to TEM- and ETG-dominated turbulence moving in electron diamagnetic direction.

Supplementary Fig. 1 suggests the turbulence to be more pronounced in the steep scenario, which is in line with the observation that all normalized kinetic gradients in the steep scenario exceed (or are equal to) those of the flat scenario, c.f. Fig. 3. For $0.4\ \mathrm{cm}^{-1} \lesssim k_\perp \lesssim 4.0\ \mathrm{cm}^{-1}$ ($0.1 \lesssim k_\perp \rho_s \lesssim 1.0$) the turbulence is ITG-dominated for both scenarios. For the smallest scales (high $k_\perp$) both cases are ETG-dominated.

**Synthetic diagnostic modeling of electron density fluctuations.** In this study the synthetic diagnostic analogue of the DBS system is the finite difference time domain code IPF-FD3D. It solves the Maxwell equations in the presence of an inhomogeneous, anisotropic (i.e. magnetized) cold plasma. The turbulent plasma density $\tilde{n}_0$ is taken directly from the GENE simulations. IPF-FD3D is run on each timestep of GENE, whereas the DBS beams are launched into the stationary plasma and the scattered wave is received and produces a single point in the DBS time series, assuming frozen turbulence. In IPF-FD3D, all microwave beams are launched in the same simulation run and discriminated in the receiver by frequency. The IPF-FD3D code generates a complex heterodyne IQ signal that is equivalent to the output of a hardware DBS system.

The fluctuation power from synthetic diagnostic modeling is is defined as the variance of the absolute IQ signal.[46] The wavenumber spectrum obtained from GENE simulations is a single-point measurement at zero radial wavenumber.

The analysis of the radial correlation lengths from IPF-FD3D is similar to the experimental analysis using the complex heterodyne signal. In contrast to experimental methods, frequency filtering is not required prior to calculating the cross-correlation function in simulations.

The electron density fluctuation time trace used for cross-phase analysis is generated by convolving the GENE turbulence field with a 2D Gaussian function and integrating the result. The radial and vertical extent of the measurement volume is set equal to that of the synthetic CECE diagnostic, as described in Ref. 11.

**Synthetic diagnostic modeling of electron temperature fluctuations.** The synthetic CECE diagnostic modeling incorporates the extended EC emission volume, estimated by ECRad and Torbeam. The procedure for extracting the time trace of the electron temperature involves convolving the turbulence field with the measurement volume, using the same approach as for the electron density in the cross-phase analysis. Similar to the experimental analysis, the synthetic CECE time traces are then correlated to obtain absolute fluctuation levels and radial correlation lengths. For the latter, standard correlation techniques are employed, as thermal noise is absent in simulations. The radial correlation length is defined as the half-width at half-maximum of Gaussian fits to the maximum of cross-correlation functions versus radial channel separation.

To determine the cross-phase, the synthetic density and temperature time traces are correlated, and the phase of the cross-power spectral density is extracted.

## Data availability

The raw data used in this study were collected by the ASDEX-Upgrade Team. The source data of kinetic profiles and the evaluated turbulence quantity data is published in Ref. 65. This link also includes the processed data shown in the publication, as well as the plotting routines for all figures of the paper. More data can be requested at the corresponding author.

## Code availability

The GENE code is available after registration at https://genecode.org/. The IPF-FD3D code is available to collaborators. Please contact C. Lechte (carsten.lechte@igvp.uni-stuttgart.de). The Torbeam code is available upon request to E. Poli (emanuele.poli@ipp.mpg.de) and upon approval by the directorate of IPP. The IDA, IDI, IDE codes are available upon reasonable request to R. Fischer (rainer.fischer@ipp.mpg.de) and upon approval by the directorate of IPP. The ECRad code is available at https://github.com/AreWeDreaming/ECRad. The ASTRA code is available upon request to G. Tardini (giovanni.tardini@ipp.mpg.de). Further evaluation routines are available upon request to the main author.

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

## Acknowledgements

We are grateful to Clemente Angioni for fruitful discussions. This work has been carried out within the framework of the EUROfusion Consortium, funded by the European Union via the Euratom Research and Training Programme (Grant Agreement No 101052200 — EUROfusion). Views and opinions expressed are however those of the author(s) only and do not necessarily reflect those of the European Union or the European Commission. Neither the European Union nor the European Commission can be held responsible for them. This work is supported by the US Department of Energy under grants DE-SC0014264, DE-SC0006419, and DE-SC0017381. The authors are grateful for the computing time provided on the Raven HPC system at the Max Planck Computing and Data Facility (MPCDF), the EUROfusion high-performance computer at CINECA (Bologna), and the GCS supercomputer HAWK at the Höchstleistungsrechenzentrum Stuttgart (www.hlrs.de) via the Gauss Centre for Supercomputing (www.gauss-centre.eu).

## Author contributions

K. Höfler played the leading role in the planning, design, and execution of the experiments, evaluated the data from the experimental turbulence measurements, supported the evaluation of other physical quantities, and coordinated the execution of the various simulation runs, including the post-processing of code results. In addition, she performed the comparison between experiment and simulation, including the integration of all contributions of the authors and participated in the writing of the paper. T. Görler performed the GENE simulations, including pre-processing tasks such as heat flux matching via gradient variation, and post-processing tasks such as evaluating raw output and generating spatially resolved time traces of fluctuation quantities and wavenumber spectra. He participated in writing the paper. T. Happel supported the experimental measurements, starting from scenario design, over choice of the measurement methods of several diagnostics, towards a wide range of post-processing of all the data included in the study. He engaged in writing the paper. C. Lechte performed the IPF-FD3D simulation runs to synthetically model DBS measurements. P. Molina did CECE measurements and supported in CECE data analysis for all turbulence quantities related to electron temperature fluctuations. M. Bergmann performed runs with the IDA code to extract profiles of $T_e$ and $n_e$. R. Bielajew supported the $n_e T_e$-cross-phase analysis. G. D. Conway supported the CECE measurements. P. David tomographically reconstructed the bolometry data to estimate the radiated power. S. S. Denk performed runs with the ECRad code to estimate the (C)ECE emission volumes. R. Fischer performed runs with the IDI and IDE codes to evaluate the profiles of $T_i$ and $v_{tor}$ and pressure constrained magnetic equilibrium. P. Hennequin supported the data analysis of DBS measurements. F. Jenko engaged in writing the paper. R.M. McDermott performed dedicated analysis of experimental $T_i$ and $v_{tor}$ profiles. A. E. White supported analysis of CECE diagnostic data. U. Stroth supported experimental scenario design, data analysis and interpretation, comparison between experiment and simulation and engaged in writing the paper. The ASDEX Upgrade Team.

## Funding

## Competing interests

The authors declare no competing interests.

## Additional information

## the ASDEX Upgrade Team

E. Alessi[8], C. Angioni[1], N. Arden[1], V. Artigues[1], M. Astrain[1], O. Asunta[9], M. Balden[1], V. Bandaru[1], A. Banon Navarro[1], M. Bauer[1], A. Bergmann[1], M. Bergmann[1], J. Bernardo[10], M. Bernert[1], A. Biancalani[11], R. Bielajew[4], R. Bilato[1], G. Birkenmeier[1,2], T. Blanken[12], V. Bobkov[1], A. Bock[1], L. Bock[1], T. Body[1], T. Bolzonella[13], N. Bonanomi[1], A. Bortolon[14], B. Böswirth[1], C. Bottereau[15], A. Bottino[1], H. van den Brand[12], M. Brenzke[16], S. Brezinsek[16], D. Brida[1], F. Brochard[17], J. Buchanan[18], A. Buhler[1], A. Burckhart[1], Y. Camenen[19], B. Cannas[20], P. Cano Megías[1], D. Carlton[1], M. Carr[18], P. Carvalho[10], C. Castaldo[21], A. Castillo Castillo[1], A. Cathey[1], M. Cavedon[22], C. Cazzaniga[13], C. Challis[18], A. Chankin[1], A. Chomiczewska[23], C. Cianfarani[21], F. Clairet[15], S. Coda[5], R. Coelho[10], J. W. Coenen[16], L. Colas[15], G. Conway[1], S. Costea[24], D. Coster[1], T. Cote[25], A. J. Creely[4], G. Croci[8], D. J. Cruz Zabala[26], G. Cseh[27], I. Cziegler[28], O. D'Arcangelo[29], A. Dal Molin[22], P. David[1], C. Day[30], M. de Baar[12], P. de Marné[1], R. Delogu[13], P. Denner[16], A. Di Siena[1], M. Dibon[1], J. J. Dominguez-Palacios Durán[26], D. Dunai[27], M. Dreval[31], M. Dunne[1], B. P. Duval[5], R. Dux[1], T. Eich[1], S. Elgeti[1], A. Encheva[32], B. Esposito[21], E. Fable[1], M. Faitsch[1], D. Fajardo Jimenez[1], U. Fantz[1], M. Farnik[33], H. Faugel[1], F. Felici[5], O. Ficker[33], A. Figueredo[10], R. Fischer[1], O. Ford[34], L. Frassinetti[35], M. Fröschle[1], G. Fuchert[34], J. C. Fuchs[1], H. Fünfgelder[1], S. Futatani[36], K. Galazka[23], J. Galdon-Quiroga[26], D. Gallart Escol'a[36], A. Gallo[15], Y. Gao[16], S. Garavaglia[8], M. Garcia Muñoz[26], B. Geiger[25], L. Giannone[1], S. Gibson[37], L. Gil[10], E. Giovannozzi[21], I. Girka[1], O. Girka[1], T. Gleiter[1], S. Glöggler[1,2], M. Gobbin[13], J. C. Gonzalez[1], J. Gonzalez Martin[26], T. Goodman[5], G. Gorini[22], T. Görler[1], D. Gradic[34], G. Granucci[8], A. Gräter[1], G. Grenfell[1], H. Greuner[1], M. Griener[1], M. Groth[9], O. Grover[1], A. Gude[1], L. Guimarais[10], S. Günter[1], D. Hachmeister[10], A. H. Hakola[38], C. Ham[18], T. Happel[1], N. den Harder[1], G. Harrer[39], J. Harrison[18], V. Hauer[30], T. Hayward-Schneider[1], B. Heinemann[1], P. Heinrich[1], T. Hellsten[6], S. Henderson[18], P. Hennequin[7], M. Herschel[1], S. Heuraux[17], A. Herrmann[1], E. Heyn[40], F. Hitzler[1,2], J. Hobirk[1], K. Höfler[34], S. Hörmann[1], J. H. Holm[41], M. Hölzl[1], C. Hopf[1], L. Horvath[28], T. Höschen[1], A. Houben[17], A. Hubbard[4], A. Huber[16], K. Hunger[1], V. Igochine[1], M. Iliasova[42], J. Illerhaus[1],

K. Insulander Björk[43], C. Ionita-Schrittwieser[24], I. Ivanova-Stanik[23], S. Jachmich[32], W. Jacob[1], N. Jaksic[1], A. Jansen van Vuuren[26], F. Jaulmes[33], F. Jenko[1], T. Jensen[41], E. Joffrin[15], A. Kallenbach[1], J. Kalis[1], A. Kappatou[1], J. Karhunen[9], C.-P. Käsemann[1], S. Kasilov[40,44], Y. Kazakov[45], A. Kendl[24], W. Kernbichler[39], E. Khilkevitch[42], M. Kircher[1], A. Kirk[18], S. Kjer Hansen[4], V. Klevarova[46], F. Klossek[1], G. Kocsis[27], M. Koleva[1], M. Komm[33], M. Kong[5], A. Krämer-Flecken[16], M. Krause[1], I. Krebs[12], A. Kreuzeder[1], K. Krieger[1], O. Kudlacek[1], D. Kulla[34], T. Kurki-Suonio[9], B. Kurzan[1], B. Labit[5], K. Lackner[1], F. Laggner[14], A. Lahtinen[9], P. Lainer[39], P. T. Lang[1], P. Lauber[1], M. Lehnen[32], L. Leppin[1], E. Lerche[1], N. Leuthold[6], L. Li[16], J. Likonen[38], O. Linder[1], H. Lindl[1], B. Lipschultz[28], Y. Liu[6], Z. Lu[1], T. Luda Di Cortemiglia[1], N. C. Luhmann[47], T. Lunt[1], A. Lyssoivan[45], T. Maceina[1], J. Madsen[41], A. Magnanimo[1], H. Maier[1], J. Mailloux[18], R. Maingi[14], O. Maj[1], E. Maljaars[12], V. Maquet[45], A. Mancini[8], A. Manhard[1], P. Mantica[8], M. Mantsinen[47,48], P. Manz[49], M. Maraschek[1], C. Marchetto[50], M. Markl[39], L. Marrelli[13], P. Martin[13], F. Matos[1], M. Mayer[1], P. J. McCarthy[51], R. McDermott[1], G. Meng[1], R. Merkel[1], A. Merle[5], H. Meyer[18], M. Michelini[1], D. Milanesio[50], V. Mitterauer[1], P. Molina Cabrera[1], M. Muraca[1], F. Nabais[10], V. Naulin[41], R. Nazikian[14], R. D. Nem[41], R. Neu[1,52], A. H. Nielsen[41], S. K. Nielsen[41], T. Nishizawa[1], M. Nocente[22], I. Novikau[1], S. Nowak[8], R. Ochoukov[1], J. Olsen[41], P. Oyola[26], O. Pan[1], G. Papp[1], A. Pau[5], G. Pautasso[1], C. Paz-Soldan[6], M. Peglau[1], E. Peluso[53], P. Petersson[35], C. Piron[13], U. Plank[1], B. Plaum[3], B. Plöckl[1], V. Plyusnin[10], G. Pokol[27,54], E. Poli[1], A. Popa[1], L. Porte[5], J. Puchmayr[1], T. Pütterich[1], L. Radovanovic[39], M. Ramisch[3], J. Rasmussen[41], G. Ratta[55], S. Ratynskaia[35], G. Raupp[1], A. Redl[44], D. Réfy[27], M. Reich[1], F. Reimold[34], D. Reiser[16], M. Reisner[1], D. Reiter[16], B. Rettino[1], T. Ribeiro[1], D. Ricci[8], R. Riedl[1], J. Riesch[1], J. F. Rivero Rodriguez[26], G. Rocchi[21], P. Rodriguez-Fernandez[4], V. Rohde[1], G. Ronchi[12], M. Rott[1], M. Rubel[35], D. A. Ryan[18], F. Ryter[1], S. Saarelma[18], M. Salewski[41], A. Salmi[9], O. Samoylov[1], L. Sanchis Sanchez[26], J. Santos[10], O. Sauter[5], G. Schall[1], A. Schlüter[1], J. Scholte[12], K. Schmid[1], O. Schmitz[25], P. A. Schneider[1], R. Schrittwieser[24], M. Schubert[1], C. Schuster[1], N. Schwarz[1], T. Schwarz-Selinger[1], J. Schweinzer[1], F. Sciortino[1], O. Seibold-Benjak[1], A. Shabbir[46], A. Shalpegin[5], S. Sharapov[18], U. Sheikh[5], A. Shevelev[42], G. Sias[20], M. Siccinio[1], B. Sieglin[1], A. Sigalov[1], A. Silva[10], C. Silva[10], D. Silvagni[1], J. Simpson[18], S. Sipilä[9], A. Snicker[9], E. Solano[55], C. Sommariva[5], C. Sozzi[8], M. Spacek[1], G. Spizzo[13], M. Spolaore[13], A. Stegmeir[1], M. Stejner[41], D. Stieglitz[1], J. Stober[1], U. Stroth[1,2], E. Strumberger[1], G. Suarez Lopez[1], W. Suttrop[1], T. Szepesi[27], B. Tál[1], T. Tala[38], W. Tang[1], G. Tardini[1], M. Tardocchi[8], D. Terranova[13], M. Teschke[1], E. Thorén[35], W. Tierens[1], D. Told[1], W. Treutterer[1], G. Trevisan[13], M. Tripský[45], P. Ulbl[1], G. Urbanczyk[1], M. Usoltseva[1], M. Valisa[13], M. Valovic[18], S. van Mulders[5,32], M. van Zeeland[6], F. Vannini[1], B. Vanovac[4], P. Varela[10], S. Varoutis[30], T. Verdier[41], G. Verdoolaege[45,46], N. Vianello[13], J. Vicente[10], T. Vierle[1], E. Viezzer[26], I. Voitsekhovitch[18], U. von Toussaint[1], D. Wagner[1], X. Wang[1], M. Weiland[1], D. Wendler[1,2], A. E. White[4], M. Willensdorfer[1], B. Wiringer[1], M. Wischmeier[1], R. Wolf[34], E. Wolfrum[1], Q. Yang[56], C. Yoo[4], Q. Yu[1], R. Zagórski[23], I. Zammuto[1], T. Zehetbauer[1], W. Zhang[56], W. Zholobenko[1], A. Zibrov[1], M. Zilker[1], C. F. B. Zimmermann[1,2], A. Zito[1], H. Zohm[1] & S. Zoletnik[27]

[8]ENEA, IFP-CNR, Milan, Italy. [9]Department of Applied Physics, Aalto University, Helsinki, Finland. [10]Instituto de Plasmas e Fusão Nuclear, Instituto Superior Técnico, Universidade de Lisboa, Lisbon, Portugal. [11]Modeling Group, Ecole supérieure d'ingénieurs Léonard-de-Vinci, Courbevoie, France. [12]Eindhoven, University of Technology, Eindhoven, Netherlands. [13]Consorzio RFX, Padova, Italy. [14]Princeton Plasma Physics Laboratory, Princeton, NJ, USA. [15]CEA/IRFM, Saint Paul Lez Durance, France. [16]Forschungszentrum, Jülich, Germany. [17]Institut Jean Lamour, Université de Lorraine, Nancy, France. [18]CCFE, Culham Science Centre, Abingdon, UK. [19]Aix-Marseille University, CNRS, Marseille, France. [20]Department of Electrical and Electronic Engineering, University of Cagliari, Cagliari, Italy. [21]ENEA, Centro Ricerche Frascati, Frascati, Italy. [22]ENEA, University of Milano-Bicocca, Milano, Italy. [23]Institute of Plasma Physics and Laser Microfusion, Warsaw, Poland. [24]ÖAW, IAP, University of Innsbruck, Innsbruck, Austria. [25]University of Wisconsin, Madison, WI, USA. [26]Universidad de Sevilla, Sevilla, Spain. [27]Centre for Energy Research, Budapest, Hungary. [28]York Plasma Institute, University of York, York, UK. [29]ENEA Consorzio CREATE, Naples, Italy. [30]Karlsruhe Institut für Technologie, Karlsruhe, Germany. [31]Institute of Plasma Physics, National Science Center Kharkov Institute of Physics and Technology, Krakov, Ukraine. [32]ITER Organization, Saint-Paul-lez-Durance, France. [33]Institute of Plasma Physics of the CAS, Praha, Czech Republic. [34]Max-Planck-Institut für Plasmaphysik, Greifswald, Germany. [35]KTH Royal Institute of Technology, Stockholm, Sweden. [36]Barcelona Supercomputing Center, Barcelona, Spain. [37]Department of Physics, Durham University, Durham, UK. [38]VTT Technical Research Centre of Finland, Helsinki, Finland. [39]ÖAW, IAP, Vienna University of Technology, Vienna, Austria. [40]ÖAW, Graz University of Technology, Graz, Austria. [41]Department of Physics, Technical University of Denmark, Kgs, Lyngby, Denmark. [42]Ioffe Institute, St, Petersburg, Russian Federation. [43]Department of Physics, Chalmers University of Technology, Gothenburg, Sweden. [44]Universitá degli Studi della Tuscia, DEIM Department, Viterbo, Italy. [45]ERM/KMS, Brussels, Belgium. [46]Ghent University, Ghent, Belgium. [47]Electrical and Computer Engineering, University of California, Davis, CA, United States of America. [48]ICREA, Barcelona, Spain. [49]Universität Greifswald, Greifswald, Germany. [50]ISC-CNR and Politecnico di Torino, Torino, Italy. [51]School of Physics, University College Cork, Cork, Ireland. [52]Technische Universität München, Garching, Germany. [53]Department of Industrial Engineering, University of Rome, Rome, Italy. [54]Budapest University of Technology and Economics, Budapest, Hungary. [55]Laboratorio Nacional de Fusión, CIEMAT, Madrid, Spain. [56]Chinese Academy of Sciences, Hefei, China.

