## [Transparent Peer Review file · Nature Communications]

Milestone in predicting core plasma turbulence: Successful multi-channel validation of the gyrokinetic code GENE

Corresponding Author: Dr Klara Höfler

Version 0:

Reviewer comments:

Reviewer #1

(Remarks to the Author)

The manuscript describes some simulation results of cyclotron dynamics in turbulence and micro-scale parameters, showing a good agreement with the experiment. From the progress of simulation research, the manuscript can be considered for publication, but the following issues must be clarified:

1. In the abstract and summary, the author refers to "an unprecedented number of simultaneous plasma observables" and "an unprecedented number of measured background and background. Fluctuation parameters were compared with simulation results." But in the article, only see the comparative analysis of two shots, how to understand "an unprecedented number of" ?
2. The author has repeatedly stated that the relevant results will play an important role for the simulation program in the prediction of macroscopic parameters such as transport and fusion device design. However, only the results of turbulence simulation are presented in this paper, and the results of transport parameters are not shown. Therefore, should the application expansion of the simulation program be carefully inferred, or should the corresponding evidence of macroscopic parameter simulation results be provided?
3. What method is used to fit the simulation in Figure 6(a)? The original simulation results are in good agreement with the experimental trend, but the simulation fitting obviously reduces the gap with the experiment too much. Refitting is recommended.
4. Can the author explain in detail why the apparent difference in the cross-correlation between the steep and flat two discharges shown experimentally (Fig.6(b)) is almost the same in the simulation?

(Remarks on code availability)

Reviewer #2

(Remarks to the Author)

Turbulence is the main cause of energy transport in the core of tokamak plasmas. This manuscript presents comparisons between experimental measurements and simulation results for a significant number of physical quantities in two plasma scenarios (called "steep" and "flat") of ASDEX Upgrade. The authors conclude (and this is the main message of the manuscript) that state-of-the-art plasma turbulence codes correctly predict core plasma transport and that, consequently, one can rely on these codes for the design of tokamak power plants.

In general, the manuscript is clearly written, and provides a careful comparison of a variety of experimentally measured quantities with predictions from turbulence simulations. This kind of work is very valuable for the community and should be published somewhere, but the manuscript does not present enough evidence for its main claim (i.e. that the the question in the title of the manuscript has an affirmative answer), and, at least in its present form, I cannot recommend it for publication in Nature Communications. I include my comments below.

Major comments:

- My main criticism has to do with the fact the authors only simulate Q_i and Q_e at a single radial location for each scenario

(see Fig. 3). I do not know the reason for this, but for the aforementioned main claim of the manuscript to be justified, a comparison (experiment vs simulation) of the radial profiles of Q_i and Q_e is needed.

– On page 8, discussing Figs. 4(c) and (d), the authors say that the synthetic diagnostic spectra agree in shape with their experimental counterparts. However, they do not discuss the reason for what (unless I am missing something) seems to be a strong quantitative disagreement. If all or part of this has to do with the experimental diagnostic not allowing to give absolute values for the density fluctuations, it should be clearly explained.

Minor comments:

– Sometimes, "el." is used for "electron", which is probably not the best abbreviation.

– Note that ρ^* , ρ_i , ρ and ρ_{pol} are used along the paper. Perhaps it would be advisable not to use "rho" for all of them.

– One page 6, it is said that "Due to the small gyroradius-to-machine-size ratio...". It took me a while to realize that the values of the gyroradius-to-machine-size ratio are given in the table on page 19. I suggest that the authors refer to the table in the main text.

– The value of k_{perp} in cm^{-1} is not particularly useful. I suggest to give $k_{perp} \rho_i$.

– In the caption of Fig. 4, ".Experimental" should be ". Experimental".

– In the last line of page 7, I think that "heat flux" should be "electron heat flux".

(Remarks on code availability)

Reviewer #3

(Remarks to the Author)

This paper claims that gyrokinetic simulation provides a reliable tool for predicting the plasma behavior in a fusion power plant. The claim is supported by a well-coordinated validation study. At the present time the gyrokinetic model is widely believed to have been validated for the transport of tokamak core plasmas, and this work contains by far the most comprehensive validation effort in terms of the number of observables that are compared.

Does this work, together with previous similar validation studies, establish gyrokinetic simulation as a reliable tool for predicting plasma behaviors in fusion power plants? The answer depends on what one means by predicting plasma behavior. What is validated in the paper is a specific theoretical model, namely, gradient-driven local gyrokinetic simulation. The model is typically used in the following way. Given a specification of the macroscopic core plasma conditions (equilibrium magnetic field, density profile, temperature profile, etc.), it is used to calculate the transport coefficients, hence confinement time. What is most needed in designing a power plant, on the other hand, is the prediction of the macroscopic plasma parameters (e.g the ion temperature profile), given the external magnetic field and external heating power, etc. To be sure, integrated simulation models that combine the local gyrokinetic model (validated in this work) with fluid transport models have been proposed to predict the temperature profile, but such models are currently under investigation and far from being validated. Coupling between the core plasma and the boundary plasma will be an important component in such integrated modeling. While the gyrokinetic model is still believed to be the favored tool for modeling transport in the boundary plasmas, the local gyrokinetic model is no longer applicable. Validation of the local gyrokinetic model in the core does not validate gyrokinetic modeling in general. The latter is vastly more complicated.

The main conclusion of the paper would therefore foster a confidence in gyrokinetic simulation, as a tool for designing fusion power plants, that is not fully supported by the provided evidences.

(Remarks on code availability)

The GENE code is well-known and widely accessible. I have not used the code myself, but it is used by some colleagues in my research group.

Version 1:

Reviewer comments:

Reviewer #1

(Remarks to the Author)

This manuscript better answers most of the reviewer's questions and may be considered for publication. However, the title of the article should be modified to a more consistent topic, such as a new turbulence simulation method. After all, the relevant content of the article is a long distance from fusion power plants and Milestone.

(Remarks on code availability)

Reviewer #2

(Remarks to the Author)

The changes made by the authors have improved the manuscript. However, my first major comment remains, and I cannot recommend the manuscript for publication in Nature Communications.

(Remarks on code availability)

Reviewer #3

(Remarks to the Author)

The new title and statements of the main conclusions now properly reflect the contributions of the article. Given the importance of the subject to the magnetic confinement fusion program, and the broad scope of this study, I recommend publication in Nature Communications.

(Remarks on code availability)

Response to the comments of Reviewer 1

November 1, 2024

The authors would like to express their gratitude to the referee for taking the time to thoroughly review the manuscript and for providing valuable comments and recommendations that have significantly improved its quality. We highly appreciate the constructive feedback, and all comments have been carefully considered. Detailed responses to each point are provided below, with changes to the manuscript highlighted in purple for clarity.

Remarks from the Reviewer to the Author

Overall comment from the Reviewer *The manuscript describes some simulation results of cyclotron dynamics in turbulence and micro-scale parameters, showing a good agreement with the experiment. From the progress of simulation research, the manuscript can be considered for publication, but the following issues must be clarified:*

We appreciate the reviewer's positive assessment of our manuscript and welcome the opportunity to clarify the issues raised.

Answers to particular comments from the Reviewer

1. *In the abstract and summary, the author refers to "an unprecedented number of simultaneous plasma observables" and "an unprecedented number of measured background and background. Fluctuation parameters were compared with simulation results." But in the article, only see the comparative analysis of two shots, how to understand "an unprecedented number of" ?*

A key achievement of this study was the simultaneous measurement of the largest number of different plasma parameters ($n=7$) in a single

discharge and the subsequent comparison with simulation results (c.f. Figure 1 left): fluctuation amplitudes in electron density and electron temperature domain, radial correlation lengths in electron density and electron temperature domain, cross-phase between electron density and temperature fluctuations, and heat flux in electron and ion domain. Indeed, it is not the number of plasma scenarios that is exceptionally high.

This point has been clarified in the manuscript:

In this work, an unprecedented number of measured background and fluctuation parameters were compared with simulation results **in two different plasma scenarios**. The high level of agreement in all parameters is impressive. The fact that agreement was achieved in **both** plasma scenarios, with flat and steep electron temperature gradients, further strengthens the positive outcome of this study.

- 2. The author has repeatedly stated that the relevant results will play an important role for the simulation program in the prediction of macroscopic parameters such as transport and fusion device design. However, only the results of turbulence simulation are presented in this paper, and the results of transport parameters are not shown. Therefore, should the application expansion of the simulation program be carefully inferred, or should the corresponding evidence of macroscopic parameter simulation results be provided?*

We thank the reviewer for pointing this out. Our work aimed at finding evidence whether the physical models used in GENE are correct. This objective cannot be reached by comparing simulated with experimental transport fluxes only, which is frequently done. Also, comparing simulated with experimental profiles by coupling transport with turbulence codes or employing flux-driven full-f codes does not cover as many aspects of the nature of turbulence as presented here. Our study compares microscopic fluctuation parameters which are much more directly related to the models used. We have revised the beginning of the manuscript to provide a more accurate and informative summary of its contents.

The title has been changed to:

Towards fusion power plants: Milestone in predicting core plasma turbulence

The abstract has been modified:

The present study addresses this important question by means of careful comparisons between state-of-the-art gyrokinetic **turbulence** simulations with the GENE code and experimental observations in the ASDEX Upgrade tokamak for an unprecedented number of simultaneous plasma observables.

The end of the introduction has been adapted:

We conclude that gyrokinetic codes have reached a high level of maturity, which allows them to be used as reliable tools to predict core plasma turbulence. This progress significantly contributes to advancing the design of future fusion power plants.

The first paragraph of the summary has been adapted:

For the prediction of core **turbulence**, gyrokinetic codes are the favored tools. They are also used to derive transport coefficients for integrated modelling [54,55] of entire plasma discharges.

The following sentence in the summary section has been removed:

The results of this study provide a positive answer to the question posed in the title of this article.

3. *What method is used to fit the simulation in Figure 6(a)? The original simulation results are in good agreement with the experimental trend, but the simulation fitting obviously reduces the gap with the experiment too much. Refitting is recommended.*

The data fitting was done in Python using `curve_fit` from `scipy.optimize`. By default, this method uses a $\sigma = 1$ for all data points. This gives the points with larger correlation lengths (at small k_{\perp}) more weight for the fit, however, which is not intended. The plot was refitted, using the same relative sigmas for all points and thus giving all measurements the same weight (regardless of the k_{\perp}). Thank you for pointing this out.

4. *Can the author explain in detail why the apparent difference in the cross-correlation between the steep and flat two discharges shown experimentally (Fig.6(b)) is almost the same in the simulation?*

Indeed, in the experiment, the correlation length of electron temperature fluctuations is larger in the steep scenario than in the flat scenario. However, in the gyrokinetic GENE simulations (involving synthetic diagnostic modelling), the correlation lengths are approximately the same. Our interpretation of this behavior is as follows.

The GENE runs are performed based on the measured kinetic profiles. This input data is varied within the experimental uncertainties to obtain the best match between the heat fluxes of electrons and ions, Q_e and Q_i , between experiment and simulation. Figure 3(g) shows that the electron heat flux in the flat scenario is matched between the simulation and the experiment. Still the simulated value exceeds the experimental value. At the same time, we observe a slightly larger value of the correlation length of the electron temperature fluctuations in the simulation. Therefore, it can be anticipated that achieving a closer match of the heat flux – though this would require substantial additional computational resources – would also result in decreased simulated correlation lengths. This would lead to the following scenario in Figure 6b: both the experimental and simulated correlation lengths indicate that the steep scenario has a slightly larger correlation length compared to the flat scenario.

Response to the comments of Reviewer 2

November 1, 2024

The authors would like to express their gratitude to the referee for taking the time to thoroughly review the manuscript and for providing valuable comments and recommendations that have significantly improved its quality. We highly appreciate the constructive feedback, and all comments have been carefully considered. Detailed responses to each point are provided below, with changes to the manuscript highlighted in purple for clarity.

Remarks from the Reviewer to the Author

Overall comment from the Reviewer *Turbulence is the main cause of energy transport in the core of tokamak plasmas. This manuscript presents comparisons between experimental measurements and simulation results for a significant number of physical quantities in two plasma scenarios (called "steep" and "flat") of ASDEX Upgrade. The authors conclude (and this is the main message of the manuscript) that state-of-the-art plasma turbulence codes correctly predict core plasma transport and that, consequently, one can rely on these codes for the design of tokamak power plants.*

In general, the manuscript is clearly written, and provides a careful comparison of a variety of experimentally measured quantities with predictions from turbulence simulations. This kind of work is very valuable for the community and should be published somewhere, but the manuscript does not present enough evidence for its main claim (i.e. that the the question in the title of the manuscript has an affirmative answer), and, at least in its present form, I cannot recommend it for publication in Nature Communications. I include my comments below.

Thank you for your positive assessment of our research. We understand and acknowledge your concerns regarding the alignment of our paper with

its title, as well as the clarity of the main message outlined in the abstract and introduction. We therefore implemented several changes.

First, we made the title more concise:

Towards fusion power plants: Milestone in predicting core plasma turbulence

We also adapted the abstract:

The present study addresses this important question by means of careful comparisons between state-of-the-art gyrokinetic turbulence simulations with the GENE code and experimental observations in the ASDEX Upgrade tokamak for an unprecedented number of simultaneous plasma observables.

In the fourth paragraph of the introduction, the following sentences have been added:

The plasma scenarios were carefully designed, and the measurement diagnostics were pushed to their limits to gather comprehensive data at two radial positions in the core plasma. Since both positions yielded equally good results, only one is presented for clarity. While validating boundary plasma models is also essential, the remarkable agreement between our measurements and simulations in the core plasma represents a significant achievement and a key step towards advancing fusion power plant design.

The end of the introduction has also been adapted:

We conclude that gyrokinetic codes have reached a high level of maturity, which allows them to be used as reliable tools to predict core plasma turbulence. This progress significantly contributes to advancing the design of future fusion power plants.

The following sentence in the summary section has been removed:

The results of this study provide a positive answer to the question posed in the title of this article.

Major comments

1. *My main criticism has to do with the fact the authors only simulate Q_i and Q_e at a single radial location for each scenario (see Fig. 3). I do not know the reason for this, but for the aforementioned main claim of the manuscript to be justified, a comparison (experiment vs simulation) of the radial profiles of Q_i and Q_e is needed.*

We agree that extending the comparison to include all turbulence quantities and heat fluxes across the entire radial profile would further enhance confidence in the presented gyrokinetic simulations. However, for technical diagnostic reasons, probing turbulence in both O-mode and X-mode DBS together with CECE is only possible within a narrow radial region, depending on the plasma scenario. To enable the simultaneous measurements of these three diagnostics, the two scenarios were carefully designed to meet all requirements and still push the operable range (of the O-mode DBS) to the limit. The simultaneous comparison of a large number of turbulence characteristics represents a significant and highly non-trivial achievement, even if only in a limited radial region. At the ASDEX Upgrade tokamak we have the opportunity to measure all these parameters simultaneously, which is very unique compared to similar machines worldwide.

The full analysis and comparison between experiment and simulation as presented in this paper has been done for two different radii: $\rho_{pol} = 0.74$ and 0.79 . Since both radial positions indicate equally good agreement in the comparison, for the sake of brevity only one radius is shown here ($\rho_{pol} = 0.79$). For the comparison of electron and ion heat flux between experimental and simulation data, we are not limited by constraints to a narrow radial region. However, comparing these without incorporating turbulence data does not enhance confidence in the simulation codes (c.f. Figure 1 left: Validation studies on single parameters have already been done in the past).

To reflect these points in the manuscript, we have added the following text to the fourth paragraph of the introduction:

The plasma scenarios were carefully designed, and the measurement diagnostics were pushed to their limits to gather comprehensive data at two radial positions in the core plasma. Since both positions yielded equally good results, only one is presented for clarity. While validating boundary plasma models is also essential, the remarkable agreement between our measurements and simulations in the core plasma represents a significant achievement and a key step towards advancing fusion power plant design.

The first paragraph in the summary was changed:

For the prediction of core turbulence, gyrokinetic codes are the

avored tools. They are also used to derive transport coefficients for integrated modelling [54,55] of entire plasma discharges.

2. – *On page 8, discussing Figs. 4(c) and (d), the authors say that the synthetic diagnostic spectra agree in shape with their experimental counterparts. However, they do not discuss the reason for what (unless I am missing somethings) seems to be a strong quantitative disagreement. If all or part of this has to do with the experimental diagnostic not allowing to give absolute values for the density fluctuations, it should be clearly explained.*

It is challenging, if not impossible, to measure absolute fluctuation amplitudes using a Doppler backscattering diagnostic. We therefore follow the common practice in our field of comparing relative fluctuation amplitudes between different scenarios or between experiment and simulation, rather than measuring absolute values. To indicate this in the plot, we’ve used an arrow labeled “arbitrary offsets.” For improved clarity regarding the measurement of relative fluctuation amplitudes, we have revised the text in the manuscript accordingly.

Turbulence was probed in O-mode **by two DBS systems** (Fig. 4(c)) and X-mode (Fig. 4(d)) polarization **by one DBS system**. For better visibility, individual vertical offsets were applied to the spectra pairs, **since the applied technique only allows for the measurement of relative fluctuation amplitudes. The offsets between the flat and steep scenarios in each spectral pair, however, are deterministic.**

Minor comments

1. *Sometimes, "el." is used for "electron", which is probably not the best abbreviation.* Thank you for pointing this out. We now use the word "electron" and changed the labels in Figure 1 accordingly.
2. *Note that ρ^* , ρ_i , ρ and ρ_{pol} are used along the paper. Perhaps it would be advisable not to use "rho" for all of them.* We see the reviewer’s point. We would like to stick to our community’s convention to use ρ for all of these parameters. However, to make the paper more accessible to readers from different backgrounds, we’ve simplified the use of parameters labeled "rho." Specifically, we’ve combined ρ and

ρ_{pol} , using only ρ throughout. Additionally, instead of referring to the ion gyro radius as ρ_i , we now simply call it "ion gyro radius" to avoid confusion. Furthermore, the text

$$\rho^* = \rho_i/a \text{ } (\rho_i: \text{ion gyroradius, } a: \text{minor plasma radius})$$

has been replaced by ρ^* .

3. *One page 6, it is said that "Due to the small gyroradius-to-machine-size ratio...". It took me a while to realize that the values of the gyroradius-to-machine-size ratio are given in the table on page 19. I suggest that the authors refer to the table in the main text.*

We have added a corresponding pointer in the manuscript:

Due to the small gyroradius-to-machine-size ratio at the radial positions of interest (c.f. Tab. 1), the latter is employed in the present cases to benefit from the higher accuracy of the underlying spectral methods.

4. *The value of k_{perp} in cm^{-1} is not particularly useful. I suggest to give $k_{perp} \rho_i$.*

We are familiar with both notations: k_{perp} in cm^{-1} , which is commonly used by experimentalists and diagnosticians, whereas $k_{perp} \rho_i$ is mostly used by theoreticians. Since this study is a collaboration between experimentalists and theoreticians, we see the pros and cons for both types of notation. In the present study, we have chosen to follow the commonly used notation by the majority of co-authors in previous publications: Hennequin, PPCF, 2004; Happel, PPCF, 2017; Lechte, PPCF, 2017. We see the importance of including a remark on the hybrid larmor radius ρ_L in the plot, where $\rho_L = \frac{\sqrt{T_e m_i}}{eB}$ with T_e , m_i , e , B referring, respectively, to the electron temperature, ion mass, elementary charge, and magnetic field. The position of $k_{perp} \rho_L = 1$ is marked with a colored line.

5. *In the caption of Fig. 4, ".Experimental" should be ". Experimental".*

Thank you for pointing out this typo; we corrected it in the manuscript.

6. *In the last line of page 7, I think that "heat flux" should be "electron heat flux".*

We clarified this point in the manuscript:

With matched surface-integrated heat fluxes in both the ion and the electron domain, the analysis now turns to the comparison of code results with experimental fluctuations measurements.

Response to the comments of Reviewer 3

November 1, 2024

The authors would like to express their gratitude to the referee for taking the time to thoroughly review the manuscript and for providing valuable comments and recommendations that have significantly improved its quality. We highly appreciate the constructive feedback, and all comments have been carefully considered. Detailed responses to each point are provided below, with changes to the manuscript highlighted in purple for clarity.

Remarks from the Reviewer to the Author

This paper claims that gyrokinetic simulation provides a reliable tool for predicting the plasma behavior in a fusion power plant. The claim is supported by a well-coordinated validation study. At the present time the gyrokinetic model is widely believed to have been validated for the transport of tokamak core plasmas, and this work contains by far the most comprehensive validation effort in terms of the number of observables that are compared.

Does this work, together with previous similar validation studies, establish gyrokinetic simulation as a reliable tool for predicting plasma behaviors in fusion power plants? The answer depends on what one means by predicting plasma behavior. What is validated in the paper is a specific theoretical model, namely, gradient-driven local gyrokinetic simulation. The model is typically used in the following way. Given a specification of the macroscopic core plasma conditions (equilibrium magnetic field, density profile, temperature profile, etc.), it is used to calculate the transport coefficients, hence confinement time. What is most needed in designing a power plant, on the other hand, is the prediction of the macroscopic plasma parameters (e.g the ion temperature profile), given the external magnetic field and external heating power, etc. To be sure, integrated simulation models that combine the

local gyrokinetic model (validated in this work) with fluid transport models have been proposed to predict the temperature profile, but such models are currently under investigation and far from being validated. Coupling between the core plasma and the boundary plasma will be an important component in such integrated modeling. While the gyrokinetic model is still believed to be the favored tool for modeling transport in the boundary plasmas, the local gyrokinetic model is no longer applicable. Validation of the local gyrokinetic model in the core does not validate gyrokinetic modeling in general. The latter is vastly more complicated.

The main conclusion of the paper would therefore foster a confidence in gyrokinetic simulation, as a tool for designing fusion power plants, that is not fully supported by the provided evidences.

We appreciate the reviewer's recognition of our study as being by far the most comprehensive validation effort to date in terms of the number of observables. This is a crucial and highly demanding test of the validity of gyrokinetics, as each additional observable significantly raises the rigor of the evaluation. The strong agreement between the experimental measurements and the simulation results reinforces our confidence in the accuracy of the gyrokinetic model employed. Nevertheless, we fully concur with the reviewer's suggestion that additional validation, particularly in the edge and divertor regions, is necessary to achieve a more complete predictive capability. Accordingly, we have revised the title and made relevant updates to the manuscript text.

New title:

Towards fusion power plants: Milestone in predicting core plasma turbulence

The following sentence in the summary section has been removed:

The results of this study provide a positive answer to the question posed in the title of this article.

In the fourth paragraph of the introduction, we added the following sentences:

The plasma scenarios were carefully designed, and the measurement diagnostics were pushed to their limits to gather comprehensive data at two radial positions in the core plasma. Since both positions yielded equally good results, only one is presented for clarity. While validating boundary plasma models is also essential, the remarkable agreement between our measurements and simulations in the core plasma represents a significant achievement and a key step towards advancing fusion

power plant design.

We also changed the end of the introduction:

We conclude that gyrokinetic codes have reached a high level of maturity, which allows them to be used as reliable tools to predict core plasma turbulence. This progress significantly contributes to advancing the design of future fusion power plants.

Finally, we changed the first paragraph of the summary section:

For the prediction of core turbulence, gyrokinetic codes are the favored tools. They are also used to derive transport coefficients for integrated modelling [54,55] of entire plasma discharges.

Remarks from the Reviewer on code availability

The GENE code is well-known and widely accessible. I have not used the code myself, but it is used by some colleagues in my research group.

Thank you for pointing this out. Access to the GENE code can be requested at the following webpage: <https://genecode.org>

Response to the comments of Reviewer 1

January 19, 2025

The authors would like to again express their gratitude to the referee for taking the time to thoroughly review the manuscript for the second time. We highly appreciate the constructive feedback, and will carefully propose a new title, as suggested by the referee. All changes to the manuscript are highlighted in purple for clarity.

Remarks from the Reviewer to the Author

Overall comment from the Reviewer *This manuscript better answers most of the reviewer's questions and may be considered for publication. However, the title of the article should be modified to a more consistent topic, such as a new turbulence simulation method. After all, the relevant content of the article is a long distance from fusion power plants and Milestone.*

Thank you for the positive assessment concerning publication. We have reconsidered the content of the paper and its connection to the title.

While acknowledging that various challenges remain on the path to a fusion power plant, we see our research tackling one of the major points: The energy confinement time of a magnetic confinement fusion device is still the quantity, which needs optimization. However, to avoid confusion in the interested readership, we follow the reviewer's proposal and choose not to include the term "fusion power plant" in the title.

Instead, we would believe that the word "milestone" accurately captures the significance of our achievement. After a quarter century of dedicated effort, we have reached a major milestone: the development of a gyrokinetic model, specifically the GENE code, which can accurately simulate the behavior of modern fusion devices. This achievement highlights the progress of

the community and in our opinion warrants the designation of a milestone in our research field.

We have changed the title from "Towards fusion power plants: Milestone in predicting core plasma turbulence" to: **Milestone in predicting core plasma turbulence: Successful multi-channel validation of the gyrokinetic code GENE**

Response to the comments of Reviewer 2

January 19, 2025

The authors would like to again express their gratitude to the referee for taking the time to thoroughly review the manuscript for the second time. We highly appreciate the constructive feedback, and will carefully propose a new title, as suggested by the referee. All changes to the manuscript are highlighted in purple for clarity.

Remarks from the Reviewer to the Author

Overall comment from the Reviewer *The changes made by the authors have improved the manuscript. However, my first major comment remains, and I cannot recommend the manuscript for publication in Nature Communications.*

Your first major comment of the first revision addresses a comparison of the ion and electron heat fluxes between experiment and simulation over the full radial region.

In Figure 3 (a,c,e) of the manuscript we show profiles of the experimentally measured kinetic quantities T_e , T_i and n_e . The solid lines are fits to the quantities to calculate the gradient lengths shown in Figure 3 (b,d,f). The local GENE simulations take as input the absolute value and the gradient at a single radial location. Therefore, the simulated heat fluxes in Figure 3 (g,h) are only obtained at a single radial location. The experimental heat fluxes in Figure 3 (g,h) are calculated using the ASTRA code, which directly solves the power balance equation. To do this, ASTRA takes the kinetic profiles from Figure 3 (a,c,e) as input and uses different models to calculate the radial location of ECRH heating, NBI heating and ohmic heating. ASTRA thus provides a complete radial profile of Q_e and Q_i . We therefore compare

single radial points of Q_e and Q_i from the simulation with a full radial profile of Q_e and Q_i from the experiment.

In order to obtain a radial profile of the GENE simulated heat fluxes, one would either have to perform many more local GENE runs, or perform global GENE runs. Both options are computationally very expensive and have been considered in the context of this work. As part of the comprehensive study, of which only the highlights are presented in this manuscript, experimental turbulence measurements at two different radial locations were compared with two local GENE simulations at the respective radii. At both radial positions Q_e and Q_i the simulation and experiment agree within the measurement uncertainties. For the sake of brevity, this paper only deals with the comparison of one radial position. Details to the full comprehensive study can be found in <https://mediatum.ub.tum.de/1686546>.

All other turbulence quantities presented in this manuscript are compared at this same radial position as well. For technical diagnostic reasons it is not possible to measure all turbulence quantities over the full radial plasma region in the same plasma scenario. This would require a massive stack of different DBS systems and CECE radiometers probing in different frequency bands. To our knowledge there is no experiment worldwide being able to probe all this simultaneously.

We recognise that a comparison of the whole radial region is another step towards a more in-depth study. However, this is currently out of scope due to the issues mentioned above.

However, to give an outlook, we also performed measurements in the plasma edge to be able to compare with GENE. However, to be able to measure in the edge plasma, we had to design a completely different plasma scenario with a higher density. So we have to do all the work again on the simulation side (extracting kinetic profiles, running GENE simulations and all the other simulations for this new scenario). This may be on the horizon for a new project, but is beyond the scope of this study.

Response to the comments of Reviewer 3

January 19, 2025

The authors would like to express their deep gratitude to the referee for taking the time to thoroughly review the manuscript for the second time.

Remarks from the Reviewer to the Author

The new title and statements of the main conclusions now properly reflect the contributions of the article. Given the importance of the subject to the magnetic confinement fusion program, and the broad scope of this study, I recommend publication in Nature Communications.

Thank you very much for your very positive assessment and recommendation to publish the manuscript in Nature Communications.